# Phytochemical Composition and Biological Activities of *Arctium minus* (Hill) Bernh.: A Potential Candidate as Antioxidant, Enzyme Inhibitor, and Cytotoxic Agent

**DOI:** 10.3390/antiox11101852

**Published:** 2022-09-20

**Authors:** Selen İlgün, Gökçe Şeker Karatoprak, Derya Çiçek Polat, Esra Köngül Şafak, Gülsüm Yıldız, Esra Küpeli Akkol, Eduardo Sobarzo-Sánchez

**Affiliations:** 1Department of Pharmaceutical Botany, Faculty of Pharmacy, Erciyes University, Kayseri 38039, Turkey; 2Department of Pharmacognosy, Faculty of Pharmacy, Erciyes University, Kayseri 38039, Turkey; 3Department of Pharmaceutical Botany, Faculty of Pharmacy, Ankara University, Ankara 06560, Turkey; 4Department of Pharmacognosy, Faculty of Pharmacy, Van Yüzüncü Yıl University, Van 65080, Turkey; 5Department of Pharmacognosy, Faculty of Pharmacy, Gazi University, Ankara 06330, Turkey; 6Instituto de Investigación y Postgrado, Facultad de Ciencias de la Salud, Universidad Central de Chile, Santiago 1783, Chile; 7Department of Organic Chemistry, Faculty of Pharmacy, University of Santiago de Compostela, Santiago de Compostela 15782, Spain

**Keywords:** *Arctium minus*, Asteraceae, antioxidant, enzyme inhibition, cytotoxic activity

## Abstract

*Arctium minus* (Hill) Bernh. (Asteraceae), which has a wide distribution area in Turkey, is a medicinally important plant. Eighty percent methanol extracts of the leaf, flower head, and root parts of *A. minus* were prepared and their sub-fractions were obtained. Spectrophotometric and chromatographic (high-performance liquid chromatography) techniques were used to assess the phytochemical composition. The extracts were evaluated for antioxidant activity by diphenyl-2-picrylhydrazil radical (DPPH^●^), 2,2′-Azino-bis 3-ethylbenzothiazoline-6-sulfonic acid (ABTS^●+^) radical scavenging, and β-carotene linoleic acid bleaching assays. Furthermore, the extracts were subjected to α-amylase, α-glucosidase, lipoxygenase, and tyrosinase enzyme inhibition tests. The cytotoxic effects of extracts were investigated on MCF-7 and MDA-MB-231 breast cancer cell lines. The richest extract in terms of phenolic compounds was identified as the ethyl acetate sub-fraction of the root extract (364.37 ± 7.18 mg_GAE_/g_extact_). Furthermore, chlorogenic acid (8.855 ± 0.175%) and rutin (8.359 ± 0.125%) were identified as the primary components in the leaves’ ethyl acetate sub-fraction. According to all methods, it was observed that the extracts with the highest antioxidant activity were the flower and leaf ethyl acetate fractions. Additionally, ABTS radical scavenging activity of roots’ ethyl acetate sub-fraction (2.51 ± 0.09 mmol/L Trolox) was observed to be as effective as that of flower and leaf ethyl acetate fractions at 0.5 mg/mL. In the β-carotene linoleic acid bleaching assay, leaves’ methanol extract showed the highest antioxidant capacity (1422.47 ± 76.85) at 30 min. The enzyme activity data showed that α-glucosidase enzyme inhibition of leaf dichloromethane extract was moderately high, with an 87.12 ± 8.06% inhibition value. Lipoxygenase enzyme inhibition was weakly detected in all sub-fractions. Leaf methanol extract, leaf butanol, and root ethyl acetate sub-fractions showed 99% tyrosinase enzyme inhibition. Finally, it was discovered that dichloromethane extracts of leaves, roots, and flowers had high cytotoxic effects on the MDA-MB-231 cell line, with IC50 values of 21.39 ± 2.43, 13.41 ± 2.37, and 10.80 ± 1.26 µg/mL, respectively. The evaluation of the plant extracts in terms of several bioactivity tests revealed extremely positive outcomes. The data of this study, in which all parts of the plant were investigated in detail for the first time, offer promising results for future research.

## 1. Introduction

Humans have throughout history successfully developed various methods of using herbs to treat illness. As a result, herbal treatment applications based on traditional methods are widely used throughout the world. Medicinal plants and their chemical components with anti-inflammatory, free radical scavenging, and anti-diabetic properties have important therapeutic roles in various health-related complications. In this context, plant extracts obtained by traditional methods or using modern techniques are utilized in clinical applications due to their chemical content, and their therapeutic effects have been investigated [1,2]. 

Free radicals are molecules produced in the form of reactive oxygen species (ROS) in biological systems. They cause destructive and irreversible degeneration of components such as lipids, proteins, and DNA, which are the basic elements of the cell [3]. There are many factors contributing to the gradual accumulation of ROS in cells. Inflammation is an organism’s physiological response to a variety of stimuli, including microbial infections, physical damage, ultraviolet light, malignantly transformed cancer cells, and immune reactions; enormous amounts of ROS are produced during inflammation [4]. Long-term inflammatory and oxidative reactions cause chronic inflammation [5]. Chronic inflammation is a main cause of aging and serious diseases such as asthma, arthritis, inflammatory bowel diseases, bronchitis, pancreatitis, liver fibrosis, cardiovascular diseases, neurodegenerative disorders, and cancer [5]. Experimental and clinical studies have shown that chronic inflammation leads to the development of cancer. In addition, oxidative enzymes are known to play a key role in inflammation, and their relationship with cancer is being investigated in greater depth [6]. Lipoxygenases are a group of oxidative enzymes involved in the regulation of inflammatory responses. Lipoxygenases and their catalysis products have been reported to be associated with carcinogenic processes such as tumor cell proliferation, differentiation, and apoptosis [6]. Diabetes is a chronic metabolic disease characterized by high blood sugar levels [7,8]. It has been reported that there is a strong link between hyperglycemia, oxidative stress, inflammation, and the development and progression of type 2 diabetes. High blood glucose levels cause the overproduction of ROS by the electron transport chain of the mitochondria [9]. Inhibition of α-glucosidase and α-amylase and regulation of oxidative stress in the digestive system are antidiabetic mechanisms of action in the treatment of diabetes [8]. When exposed to ultraviolet radiation, human skin produces an abundance of ROS, which activates a variety of biological responses. Increasing ROS levels activate α-melanocyte-stimulating hormone in the epidermis, activating tyrosinase, and finally stimulating melanocytes to produce melanin [10]. This results in melanogenesis, referring to the over-synthesis of melanin pigments. Melanogenesis can increase tumor growth or induce tumor progression, and can also increase the risk of cancer and Parkinson’s disease [11]. Melanogenesis and antioxidant defense mechanisms have a complex and fascinating interaction. This relationship is associated with ROS scavenging activity According to the synergistic effect in the relationship, tyrosinase inhibitors increase the effectiveness of antioxidants when they are active. The resulting free radicals are cleared and melanin production is reduced. [10]. Therefore, antioxidant metabolites work to stop the overproduction or to effect the elimination of reactive oxygen species before they harm vital cellular components, preventing oxidative damage to cellular components. For this reason, research and discovery of natural antioxidant components are of great importance in the fight against diseases caused by reactive oxygen derivatives [12].

*Arctium* (Asteraceae), a Mediterranean genus with species thought to have high nutritional and medicinal value, consists of 17 species, three of which (*A. lappa* L., *A.tomentosum* Mill, and *Arctium minus* (Hill) Bernh.) are widely distributed. These biennial herbaceous species are defined as global invasive plants that can spread all over the world (especially in temperate regions of Europe and Asia, and occasionally in subtropical and tropical regions) [13,14,15,16].

*Arctium minus* is also known as “lesser burdock”, “petite bardane”, and “cibourroche”. This species’ leaves have traditionally been utilized to cure rheumatic pains, fever, sunstroke, wounds, general infections, skin and body inflammations, alopecia, and bladder diseases [17,18,19]. Furthermore, information on its traditional application to the patient’s body with vinegar or milk to stimulate perspiration has been noted [20]. Snake and scorpion bites are sometimes treated with an infusion of the roots and leaves [21]. Basal leaves and stems may be eaten raw or boiled due to their bitter taste, they are also used to stimulate liver functions and increase appetite [22]. Lignans, fatty acids, acetylenic compounds, phytosterols, polysaccharides, caffeoylquinic acid derivatives, flavonoids, terpenes, terpenoids, volatile compounds, and fatty acids are all found in *Arctium* species [16]. In addition to its anti-inflammatory properties, it also possesses anti-cancer, anti-diabetic, antioxidant, hepatoprotective, gastroprotective, antibacterial, antiviral, and anti-allergic properties [23].

Considering the studies conducted with *A. minus*, it is remarkable that the biological activities and chemical composition of the species have not been adequately studied in different organs of the plant. For this reason, the scientifically diverse biological activities of this plant, which has historically been used in the treatment of various diseases, are described in this paper for the first time. Radical scavenging and β-carotene linoleic acid bleaching assays, α-amylase, α-glucosidase, lipoxygenase, and tyrosinase enzyme inhibition tests, and MTT ((3-(4,5-Dimethylthiazol-2-yl)-2,5-Diphenyltetrazolium Bromide) assay in breast cancer cell lines were used to assess the biological activity of *A. minus* root, leaf, and flower heads. The phytochemical composition was carried out with HPLC analysis. The results can serve as a roadmap for future research on *A. minus* and provide useful information to better understand its traditional usage.

## 2. Materials and Methods

### 2.1. Plant Material 

*A. minus* was collected from Kayseri-Sarığolan (39°04′58.2″ N 35°58′36.3″ E) in 2019. After the herbarium samples were prepared, the collected and identified plants were documented and preserved at Ankara University Faculty of Pharmacy Herbarium, with the code AEF 30946.

### 2.2. Extraction Procedure

The plants were dried under suitable conditions after the flower heads, leaves, and roots had been separated. Each part of the plant was extracted by treatment with 80% methanol for three days. The extracts were lyophilized after the methanol was evaporated under vacuum. The powdered extracts were first dispersed with water to be fractionated and were then subjected to liquid–liquid extraction with dichloromethane, ethyl acetate, and *n*-butanol, respectively. All prepared sub-fractions and the leftover water sub-fraction were lyophilized after being withdrawn from their solvents.

### 2.3. Chemical Analysis

#### 2.3.1. Total Content of Phenolic Compounds

Spectrophotometric methods were employed to assess the total phenol and flavonoid content of plant extracts. The extracts were mixed with distilled water (3.95 mL), Folin-Ciocalteu reagent (250 µL), and 20% Na_2_CO_3_ (750 µL) and kept at 25 °C for 2 h before being measured at 760 nm. Gallic acid was used as a standard, and three measurements were taken in parallel [24]. 

#### 2.3.2. Total Content of Flavonoid Compounds

To determine the total flavonoid content, 4 mL of distilled water and 0.3 mL of 5% NaNO_2_ were added to the concentration-adjusted extracts. After 5 min, 0.3 mL of 10% AlCl_3_6H_2_O was added. Then, by adding 2 mL of 1M NaOH, the total volume was made up to 10 mL with distilled water. Samples were read at 510 nm using a Shimadzu Spectrophotometer UV 1800, (Washington, USA). Catechin was used as the reference substance [25]. 

#### 2.3.3. Analysis of Phenolic Acids and Flavonoids by HPLC

For HPLC analysis (Agilent 1100 series with diode array detector, New York, USA), all fractions were prepared at a concentration of 4 mg/mL and stock solutions of each standard (caffeic acid, chlorogenic acid, coumaric acid, ferulic acid, and rutin) were prepared with methanol (500 µg/mL). A Waters Spherisorb^®^ (Philadelphia, PA, USA) C18 column (25 cm × 4.6 mm, 5 µm) was used for analysis. A gradient system with a mobile phase of 0.01% formic acid (A) and acetonitrile (B), a flow rate of 1 mL/min, a column temperature of 40 °C, and a wavelength of 330 nm was used for the analysis. For the calibration curve, five different concentrations of standards were injected in triplicate. The calibration curve equation and the coefficient of correlations were determined. Accuracy, precision, limit of detection (LOD), limit of quantitation (LOQ), and recovery values were calculated for method validation [26,27]. The precision of the method (intra-day and inter-day variation) was carried out, and differences were expressed by relative standard deviation (RSD). For analysis of LOD and LOQ values, 10 injections of standards were made and signal–noise values were calculated. The LOD signal–noise value was 3:1, while the LOQ signal–noise value was 10:1. For recovery analysis, three different known concentrations of the standard were added to the sample and the recovery percentage was calculated. Minor changes were made in flow rate, column temperature, mobile phase and wavelength for the robustness analysis and it was seen that they did not affect the analysis.

### 2.4. Antioxidant Activity

#### 2.4.1. DPPH^●^ Radical Scavenging Activity

The DPPH radical scavenging effects of the samples were investigated using the Gyamfi et al. method [28]. The extracts were mixed with Tris-HCl buffer (50 nM, pH 7.4) and DPPH solution, prepared in 0.1 mM methanol. As a positive control, synthetic antioxidants such as standard antioxidants BHA and BHT (Butylated Hydroxy Anisol, Butylated Hydroxy Toluene) were used. After 30 min of room temperature incubation in the dark, the absorption spectra of the samples were recorded at 517 nm. The process was repeated three times in parallel, and inhibition % calculations were made using the following equation.
Inhibition % = [(Abs_control_ − Abs_sample_)/Abs_control_] × 100 (1)

#### 2.4.2. ABTS^●+^ Radical Scavenging Activity

The samples’ ABTS^●+^ radical scavenging effects were defined according to the method in the literature [29]. By keeping an aqueous solution of ABTS and K_2_S_2_O_8_ (2.45 mM, final concentration) in the dark for 12–16 h, an ABTS^●+^ radical (7 mM) was obtained. Its absorbance was adjusted to 0.700 (±0.030) at 734 nm. The reaction kinetics were measured and recorded at 734 nm at 1-min periods for 30 min using 990 µL of the prepared radical solution and 10 µL of the extract samples. Trolox equivalents were calculated as percentages of inhibition measured against concentration (TEAC). BHA and BHT were used as positive controls. Experiments were repeated three times in parallel, and mean values were calculated

#### 2.4.3. β-Carotene/Linoleic Acid Bleaching Inhibition Assay

The antioxidant activity of the extracts was determined according to the β-carotene bleaching method [30]. For this, 5 mg of β-carotene was dissolved in 25 mL of chloroform. An aliquot of the β-carotene solution was added to the vial containing linoleic acid (40 mg) and Tween 20 (400 mg). After the chloroform evaporated, distilled water (50 mL) was added slowly. BHA and BHT were used as positive controls. Blanks of the control and the samples were also prepared without β-carotene. The prepared emulsion and samples were subjected to thermal autoxidation by being kept at a constant temperature. The bleaching rate of β-carotene was monitored by measuring absorbance (470 nm) at 15-min intervals. The results of experiments were calculated as antioxidant activity capacity (AAC) using the following equation:AAC% = [1 − (Abs_0sampl_e − Abs_90sample_)/(Abs_0control_ − Abs_90control_)] × 100(2)

### 2.5. Enzyme Inhibitory Activity

#### 2.5.1. Lipoxygenase Inhibitory Activity

The lipoxygenase inhibitory activity of the samples was tested spectrophotometrically using a Cayman LOX Inhibitor Screening Assay Kit. Absorbance was recorded at 490 nm. Nordihydroguaiaretic acid (NDGA) was used as a positive control.

#### 2.5.2. Tyrosinase Inhibitory Activity

Tyrosinase inhibition assay was applied by making certain modifications to the method described by Chang et al. [31]. Kojic acid was used as a standard material. Kinetic readings were taken with a microplate reader at 30-s intervals to determine the linear change in absorbance at 475 nm. A 96-well microplate was used for measurements in all experiments. All samples were studied at concentrations of 100 µg/mL and 500 µg/mL. All enzyme assay measurements were repeated three times, and the results were expressed as Mean ± Standard Error Mean. IC50 values were calculated for samples showing more than 50% inhibition. Inhibition % calculations were carried out using Equation (1). 

#### 2.5.3. α-Glucosidase Inhibitory Activity

The α-glucosidase enzyme inhibitory activity assay was performed according to the method reported by Liu et al. [32]. In this process, 50 µL of 2 U/mL α-glucosidase solution was mixed with 1000 µL of phosphate buffer and 200 µL of extract/acarbose. After incubation for 10 min at 37 °C, 5 mM of 50 µL of p-nitrophenyl-α-D-glucopyranoside (pNPG) was added and the mixture was incubated again at 37 °C for 20 min. Then, 2000 µL of 0.2 M sodium carbonate and 4700 µL of distilled water were added to stop the reaction, and absorbances were measured at 405 nm using a spectrophotometer. Inhibition % calculations were made using Equation (1). 

#### 2.5.4. α-Amylase Inhibitory Activity

The inhibitory effects of the extracts on the α-amylase enzyme were investigated by the modified Sigma-Aldrich method. In a test tube, varying concentrations of 40 µL of extract/acarbose, 160 µL of 20 mM phosphate buffer (pH 6.9, containing 6.7 mM sodium chloride), and 200 µL of α-amylase enzyme solution (EC3.2.1.1, type VI, Sigma; 20 units/mL) were mixed. After incubating at 25 °C for 5 min, 400 µL of starch solution (0.5% *w*/*v*) was added as a substrate and incubated at 25 °C for another 3 min. At the end of the incubation period, 200 μL of dinitro salicylic acid reagent were added to the test tubes, which were kept in a water bath at 85 °C for 15 min. At the end of the period, all tubes were removed from the water bath. After adding 4000 μL of distilled water, their absorbance at 540 nm wavelength was measured with a spectrophotometer. Inhibition % calculations were made using Equation (1). 

### 2.6. Cytotoxic Activity 

#### 2.6.1. Cell Cultures

Cells were purchased from American Type Culture Collection (Manassas, VA, USA). MDA-MB-231 (ATCC HTB-26 Human Breast Cancer Cell Series), and MCF-7 (ATCC CCL-222, Human Breast Cancer Cell Series) were grown in Dulbecco’s Modified Eagle’s Medium (DMEM) containing 1% penicillin/streptomycin and 10% fetal bovine serum (FBS) (Gibco Invitrogen, Grand Island, NY, USA). Cell cultures were kept at 37 °C in 5% CO_2_ and 95% air.

#### 2.6.2. Determination of Cell Viability with MTT Cytotoxicity Assay

At 24 h before the study, the cells in the flask were counted and inoculated into a 96-well microplate with 1 × 10^4^ cells in 100 μL per well. After 24 h, the media on the cells that adhered to the plate were discarded. The stock solution of the extracts was prepared at 1 mg/mL. Then, the extracts were prepared by dilution in the medium at concentrations of 7.81 µg/mL, 15.6 µg/mL, 31.25 µg/mL, 62.5 µg/mL, 125 µg/mL, 250µg/mL, 500 µg/mL, 1000 µg/mL, and 100 µL of each were added to wells. At the end of 48 h, the media in the wells were emptied. The wells were filled with 100 µL of MTT solution diluted 1/10 with medium from a 5 mg/mL (in PBS) MTT stock. The plates were kept in an incubator with CO_2_ at 37 °C for 4 h. At the end of 4 h, 100 μL of DMSO was added to each well and formazan crystals formed by MTT were dissolved. At the end of 10 min, each well was read at 540 nm wavelength using a microplate reader [33].
% Viability = [(Abs_sample_ × 100)/Abs_control_]

### 2.7. Statistical Analysis

Analysis was executed in triplicate and the mean values were calculated. All the data are presented as the mean ± standard deviation (SD), relative standard deviation (RSD), linear regression analysis. Calculations were performed using the Microsoft Excel program. 

The Levene test was used to evaluate variance homogeneity. One-way analysis of variance was used for comparisons between more than two groups. The Dunnett T3 test and Tukey’s test were used for multiple comparisons. The data were evaluated with SPSS Version 11.0 statistic software package. The significance level was set at *p* <0.05. 

## 3. Results and Discussion

### 3.1. Chemical Analysis

The quantities of total phenol and flavonoids in the *A. minus* sub-fractions were calculated. Table 1 shows the total phenol and flavonoid determination results for the extracts. The ethyl acetate fraction of root (Arc R EtOAc) was found to have the highest levels of flavonoids (158.56 ± 12.87 mg_CA_/g_extract_) and total phenols (364.37 ± 7.18 mg_GAE_/g_extract_).

In a study investigating the biological activities of water and ethanol extracts of *A. minus* leaves, the total phenol content of the water extract was calculated as 58.93 ± 2.72 mg/g, and the total phenol content of the ethanol extract was calculated as 58.93 ± 2.72 mg/g [18]. In another study, the total phenol and flavonoid contents of the ethanol extract of *A. lappa* roots were found to be 19.35 mg/g and 12.29 mg/g, respectively. When compared with the results of this study, the total phenol content values of the water sub-extracts, which showed the lowest content in the leaf and root, were determined as 120.01 ± 2.82 mg_GAE_/g_extract_ and 89.47 ± 2.13 mg_GAE_/g_extract_, respectively; it was interpreted that they contained higher levels than the other studied species described in the literature [34].

The quantities of chlorogenic acid, caffeic acid, coumaric acid, ferulic acid, and rutin in the plant extracts were calculated (Figure 1 and Figure 2) and the results are shown in Table 1. Calibration values, precision data, and statistical data from the recovery assays are shown in Table 2, Table 3 and Table 4. 

The amounts of chlorogenic acid (8.855 ± 0.175%) and rutin (8.359 ± 0.125%) in the ethyl acetate sub-fraction of the leaf were found to be higher than those of the other compounds. Likewise, it was determined that the leaf butanol extract (Arc L BuOH) was rich in chlorogenic acid content (8.608 ± 0.292%). Coumaric acid was not detected in any extract except flower and leaf ethyl acetate extract (Arc F EtOAc and Arc L EtOAc). Rutin was also recorded detected only in leaf sub-fractions (excluding Arc L Aqua extracts).

Studies on the biologically active components of *Arctium* species have generally focused on *A. lappa* species, and have revealed that the biological activities of *A. lappa* are due to lignans, arctiin, arctigenin, and polysaccharides. It has also been noted that these compounds show antitumor, antibacterial, antiviral, hepatoprotective, and antiurolytic activities in combination with polyphenols (flavonoids and polyphenolcarboxylic acids) [35,36]. Studies on the phytochemical content of *A. minus* have been limited. For this reason, this research is the first detailed study on the determination of the phenolic content of plant parts from the *A. minus* species collected from Turkey. 

The literature states that rutin and isoquercetin are the two major components in ethanolic extracts of *A. minus* leaves [16]. Rutin was found to be the major component in the leaf ethyl acetate extract (Arc L EtOAc) and was calculated at 8.359 ± 0.125%. It has also been noted that caffeoylquinic acid derivatives can be found as major active ingredients in *Arctium* species, and these contribute to the plants’ extraordinary antioxidant properties [16]. In our study, chlorogenic acid was identified by the literature as the major component especially in leaf ethyl acetate and butanol sub-fractions. Studies indicate that *Arctium* species are cross-pollinating, and this creates significant differences in chemical content within and between species. In particular, *A. minus* and *A. lappa* had high variability in terms of flavonoids and hydroxycinnamic acids. *A. lappa* was particularly rich in hydroxycinnamic acids, while *A. minus* contained high levels of specific hydroxycinnamic compounds along with various flavonoid compounds. For this reason, it is of great importance to perform a content analysis of *A. minus* species [37].

### 3.2. Antioxidant Activity

The antioxidant capacities of the extracts were measured by DPPH, ABTS, and β-carotene/linoleic acid assay. Antioxidant activity was evaluated by comparing extracts with synthetic antioxidants BHT and BHA, and the results are given in Table 5.

The DPPH^•^ scavenging activation test is a widely used spectrophotometric methods to evaluate antioxidant capacity. DPPH^•^ is employed to represent the radicals in our body and the radical scavenging effects of the extracts are evaluated [38]. In the present study, EC50 values of flower, root, and leaf methanol extracts and their sub-fractions were compared, and the significance of antioxidant activity was evaluated with positive controls. In particular, the activity of ethyl acetate sub-fractions of plant parts was found to be statistically significant with positive controls for BHT and BHA (*p* < 0.01). The lowest EC50 values were observed in the dichloromethane sub-fractions of the plant. In addition, leaf and flower butanol sub-fractions showed high antioxidant activity (Arc L BuoH: 0.048 ± 0.001 mg/mL and Arc F BuOH: 0.043 ± 0.003 mg/mL), while the root butanol sub-fraction showed low activity (Arc R BuOH: 1.126 ± 0.019) (Table 5). In a study by Erdemoğlu et al., the DPPH radical scavenging effects of *A. minus* water and ethanol extracts were evaluated. The IC50 values of the extracts were calculated as 5.33 ± 0.62 and 7.18 ± 0.25 mg/mL for the water and ethanol extracts, respectively [18]. Conspicuously, for all the plant parts analyzed in the current study, lower IC50 values were calculated in water and alcohol extracts. This may be due to the chemical content of the plants studied, as well as differences in the extraction methods and experimental methods applied to the plants. In a study by Chui et al. on *A. lappa* leaves, it was determined that the plant is rich in morin and quercetin 3-*O*-rhamnoside compounds, and the antioxidant activity (DPPH: 2025.33 ± 84.15 μmol Trolox/g, ABTS: 159.14 ± 5.28 μmol Trolox/g) of the leaf flavonoids of the plant was quite high when tested in vitro [39]. *A. lappa* root extracts were examined in another study, and it was discovered that the plant’s ethyl acetate fraction had the highest levels of total phenolics, total flavonoids, caffeic acid derivatives, and phenolic acids (mostly chlorogenic, caffeic acid, and p-coumaric acids). In the hexane fraction, only triterpenes were discovered. Additionally, the ethyl acetate fraction displayed the best antioxidant activity due to its high polyphenol concentration. While the DPPH radical scavenging activity of the root ethyl acetate extract was calculated as 308.3 ± 6.1 mM Trolox/g extract, the activity in the hexane and chloroform extracts was calculated as 8.9 ± 0.1 mM Trolox/g extract and 41.1 ± 0.3 mM Trolox/g extract, respectively [40]. Similarly, the DPPH radical scavenging effect of ethyl acetate extract was higher in *A. minus* root extract (EC50: 0.042 ± 0.00 mg/mL), while the apolar dichloromethane extract showed a lower radical scavenging activity (EC50: 0.630 ± 0.056 mg/mL).

ABTS radical scavenging activity assay is another widely used method, in which ABTS is used as an oxidant and radical reduction is observed relative to absorbance before and after the addition of antioxidants to the study samples. Results from the ABTS test are given as Trolox-equivalent antioxidant capacity (TEAC) [41]. The antioxidant activities of extracts and sub-fractions of *A. minus* were evaluated by kinetic measurement at 0.5 mg/mL and 0.25 mg/mL concentrations for 30 min. The antioxidant activities of Arc R EtOAc and Arc F EtOAc extracts at 0.25 mg/mL concentration were calculated as 2.289 ± 0.014 mmol/L Trolox and 2.512 ± 0.14 mmol/L Trolox, respectively. According to the results, it was noted that these extracts were more effective than the positive controls BHT and BHA. In addition, at both concentrations it was found that the extracts exhibiting the lowest activity had water and dichloromethane sub-fractions belonging to the leaf, flower, and root. In a study investigating the ABTS radical scavenging effects of teas prepared from *A. lappa* roots by steaming, drying, and roasting methods, inhibition % was determined as 75.13 ± 1.962, 62.10 ± 1.10, and 85.66 ± 2.77, respectively [42]. In our study, the ABTS radical scavenging activity of the root water extract Arc R Aqua was calculated as 1.199 ± 0.08 mmol/L Trolox at 0.5 mg/mL concentration. In another study, the antioxidant capacities of 70% methanol extracts prepared from the roots, leaves, and seeds of *A. lappa* were measured, and the TEAC values of the extracts were found in the range of 67.39–1.63 μmol Trolox equivalent/100 g dry weight [43].

The β-carotene linoleic acid bleaching assay is a popular model for studying the oxidation of unsaturated fatty acids [44]. The change in color of β-carotene in reaction with the linoleic acid-free radical is the basis of the β-carotene/linoleic acid test. At high temperatures, this radical is formed by removing the hydrogen atom between the two double bonds of linoleic acid. Antioxidants can prevent β-carotene degradation by reacting with the free-radical linoleate or any other radical formed [45]. According to the results, no statistically significant difference was observed between Arc L MeOH, Arc R EtOAc, and Arc F EtOAc extracts, and they were found to be as protective as positive control BHT and BHA at 30 min. These extracts preserved the color and absorbance of β-carotene longer than the other extracts. When the 90-minute absorbance was evaluated, it was determined that Arc R Aqua, Arc R MeOH, Arc F Aqua, and Arc L CH_2_Cl_2_ were statistically significant among themselves, and their protective effects were quite low. At 60 and 90 min, BHT and BHA were found to be more protective than any other extract. For the first time, the antioxidant activities of extracts of the *Arctium* plant were determined by this method.

As a result, the free-radical-scavenging activity of different parts of the plant was evaluated by different methods. It was revealed that the active components of the plant were concentrated in different parts of their tissues and demonstrated significantly different free radical scavenging activity.

### 3.3. Enzyme Inhibitory Activity

Natural processes in an organism cooperate to prevent oxidative damage to cellular components, by enabling the formation or removal of reactive oxygen species by enzymes and antioxidant metabolites [46]. Hence, it is critical to evaluate the antioxidant and enzyme inhibition activities of natural compounds. Lipoxygenase plays a role in the synthesis of inflammation markers [47]. Overproduction of lipoxygenase metabolites has been associated with inflammatory diseases including cancer, myocardial rupture, post-ischemic inflammation, allergic asthma, skin diseases, metabolic syndrome, and neurodegenerative diseases such as Alzheimer’s [48]. Tyrosinase is one of the key enzymes in the biosynthetic pathway of melanin [49]. Overactivity of the tyrosinase enzyme leads to excessive production of melanin, which causes hyperpigmentation of the skin. Excessive accumulation of epidermal pigmentation may cause certain dermatological disorders associated with freckles, melasma, blemishes, and senile lentigines [50]. Furthermore, tyrosinase is linked to Parkinson’s disease and other neurodegenerative diseases [51,52].

Results of the lipoxygenase and tyrosinase enzyme inhibition activity analysis of extracts and standards are given in Table 6. According to the results, Arc F Aqua, Arc L BuOH, Arc R BuOH, Arc K Aqua, and Arc L Aqua showed weak inhibition against the Lipoxygenase enzyme, while other samples showed no inhibitory activity. In a study evaluating the antioxidant and anti-lipoxygenase activity of *A. lappa* and *A. tometosum* species, it was noted that the extracts were characterized by strong antioxidant properties but showed weak enzyme inhibition activity [53]. Similarly, weak lipoxygenase enzymeinhibition activity was observed in *A. minus* extract and its fractions.

Arc L EtOAc (99.65%) was demonstrated the most effective inhibition of the tyrosinase enzyme at 500 µg/mL concentration. The IC50 value of Arc L EtOAc (93.00 µg/mL) was found to be approximately three times the IC50 of kojic acid (30.00 µg/mL). The IC50 values of other effective fractions, Arc F BuOH, Arc R EtOAc, and Arc L MeOH were calculated as 176.67, 218.75, and 401.67 µg/mL, respectively. As a result, it was determined that Arc F BuOH, Arc R EtOAc, and Arc MeOH extracts, which are thought to be rich in phenolic compounds, have a high antioxidant capacity and in parallel show high tyrosinase enzyme inhibition activity. In a study evaluating the tyrosinase enzyme inhibition activity of methanol extract and sub-fractions of *Arctium lappa* roots, the highest activity was observed in the ethyl acetate fraction (IC50: 1.326 ± 0.158 mg/mL) [54]. In our data analysis, the IC50 value of the root ethyl acetate extract was calculated as 218.75 ± 0.83 µg/mL. However, it was observed that the leaf ethyl acetate extract (IC50: 93.00 ± 1.54 µg/mL) was more active than the root ethyl acetate extract.

Inhibition of α-glucosidase and α-amylase in the digestive system, and regulation of oxidative stress, are very important as antidiabetic mechanisms of action in the treatment of diabetes [8]. A strong link has been reported between hyperglycemia, oxidative stress caused by hyperglycemia, inflammation, and the development and progression of type 2 diabetes [9]. As inhibitors of carbohydrate hydrolyzing enzymes, α-glucosidase and α-amylase offer an effective strategy to regulate or prevent hyperglycemia by controlling starch degradation [55].

The inhibitory activity of extracts prepared from different parts of *A. minus* on α-amylase and α-glucosidase was investigated in this study for the first time, and the results are given in Table 6. Among all extracts, only the leaf extracts (excluding leaf ethyl acetate extract) showed α-amylase inhibition activity at a concentration of 1 mg/mL. In the α- glucosidase inhibition assay, the dichloromethane extract of the *A. minus* leaf had the highest enzyme inhibition activity, with 87.12% inhibition, compared with the other extracts and with acarbose at a concentration of 1 mg/mL. Nickavar and Yousefian found that crude hydroalcoholic (70% ethanol) extract of *A. lappa* root caused a 35.06 ± 0.38% reduction in α-amylase activity at a dose of 2.304 mg/mL [56]. In another study, the effects of the *A. lappa* methanol, ethyl acetate, and water extracts on α-glucosidase enzyme inhibition activity were investigated, and it was reported that ethyl acetate extract caused 77.5%, 50.3%, 43.1% inhibition in a dose-dependent manner at 0.2, 0.1 and 0.002 mg/mL doses, respectively [57].

Synthetic α-glucosidase inhibitors have been developed, such as acarbose, miglitol, and voglibose, but these compounds have been associated with certain serious side effects, including gastrointestinal problems [58]. Therefore, other studies have focused on the identification of natural α-glucosidase inhibitors. Recent research has shown that fructoioigosaccharides and bioactive polyphenols from burdock, especially chlorogenic acid, exhibit antidiabetic effects [59]. In addition, evidence from in vitro experimental studies indicates that phenolic compounds inhibit α-amylase and α-glucosidase activities. Chui et al. investigated the antihyperglycemic effects of the flavonoids of *A. lappa* leaves and reported promising results (α-amylase IC50 value of 92.01 μg mL^−1^ and α-glucosidase IC50 value of 29.49 μg mL^−1^) [60]. Interestingly, in our study it was observed that Arc L EtOAc extract, which had the highest total phenol and flavonoid content, did not show any activity.

### 3.4. Cytotoxic Activity

Breast cancer remains the main cause of death in women worldwide. Therefore, it is very important to identify alternative or new drug targets and to better understand the metabolic adaptations of various breast cancer subtypes. Breast cancer is classified as hormone-sensitive ER/PR+, representing early benign tumor status, or aggressive ER/PR-, representing late-stage metastasis. In this study, MCF-7 cells with ER/PR+ and MDA-MB-231 cells with ER/PR- were selected as models for these two types of breast cancer [61].

The effects of *A. minus* extracts on MCF-7 and MDA-MB-231 cell lines were observed using the MTT method. The results are given in Table 7. When all extracts were tested for cytotoxicity, the dichloromethane fractions of the roots, leaves, and flowers were found to be the most effective, particularly in the MDA-MB-231 cell line. The IC50 value of Arc F CH_2_Cl_2_ extract was calculated as 10.80 ± 1.26 µg/mL. The IC50 values of Arc R CH_2_Cl_2_ and Arc L CH_2_Cl_2_ extracts were 13.41 ± 2.37 and 21.39 ± 2.43 µg/mL, respectively, in the MDA-MB-231 cell line. Similarly, flower, root, and leaf dichloromethane extracts were found to be more toxic than other extracts in the MCF-7 cell line. The extracts with the weakest activity were found to be Arc L Aqua, Arc R MeOH, and Arc R Aqua (IC50 > 1000). Interestingly, Arc L EtOAc, Arc F EtOAc, and Arc R EtOAc extracts were found ineffective in terms of cytotoxic activity, despite having very high antioxidant activity.

Previously, the cytotoxic effects of *A. lappa* have been reported in the literature. Ghafari et al. evaluated the apoptotic and necrotic effects of extracts obtained from *A. lappa* roots, using MCF-7 and MDA-MB-231 cell lines, and reported that the extracts had a high antiproliferative effect even at 10 µg/mL concentration [62]. In other studies, it was discovered that aqueous extracts of *A. lappa* roots had no cytotoxic effect on the MCF-7 cell line, whereas the root dichloromethane extract of *A. lappa* had a strong cytotoxic effect [63,64]. According to our research, it was determined that water sub-extracts of *A. minus* showed a weak cytotoxic effect, and dichloromethane sub-extracts had a very significant cytotoxic effect compared with other extracts.

Arctigenin, which is isolated from *A. lappa* and is one of the major components of *Arctium* species, has been proven in studies to have strong anticancer effects [65]. Generally, petroleum ether, dichloromethane, and butanol extracts of the plant have been reported to be rich in this compound [66]. These data may explain the stronger cytotoxic effect of *A. minus* dichloromethane extracts compared with that of other extracts. In addition, other research discovered that the arctigenin compound significantly reduced cell proliferation when applied in combination with chlorogenic acid and cinnamaldehyde to MCF-7 and MDA-MB-231 cells [67]. These results also indicate that the compounds in the extracts can exhibit stronger biological activities by providing synergistic effects together.

## 4. Conclusions

In the present study, the biological activities of the methanol extract prepared from the leaves, roots, and flowers of *A. minus* were investigated in detail, as were the dichloromethane, ethyl acetate, butanol, and water sub-fractions of these extracts. To the best of our knowledge, this is the first detailed bioactivity assessment study to examine different parts of *A. minus* plants. Antioxidant, cytotoxic, α-amylase, α-glucosidase, lipoxygenase, and tyrosinase enzyme inhibition activities of 15 different extracts were evaluated, and phytochemical analyses were carried out. It was determined that the extracts with high antioxidant effects were rich in phenolic content. In addition, it was observed that the root, flower, and leaf ethyl acetate extracts of the plant showed high antioxidant activity. The plant’s sub-fractions of dichloromethane were found to be moderately efficient against the two primary kinds of breast cancer, MCF7 (ER/PR-) and MDA-MB-231 (ER/PR+). The extract with the highest α-glucosidase enzyme inhibitory activity was identified as Arc L CH_2_Cl_2_. Arc R EtOAc and Arc L EtOAc sub-fractions each showed strong tyrosinase enzyme inhibition activity.

This research provides thorough information that might serve as a scientific foundation for using *Arctium minus* as a source of bioactive chemicals with diverse roles in the nutraceutical and pharmaceutical sectors. Further study should be undertaken in the future to address a variety of concerns including bioavailability, formula ability, and toxicity characteristics of the extracts examined.

## Figures and Tables

**Figure 1 antioxidants-11-01852-f001:**
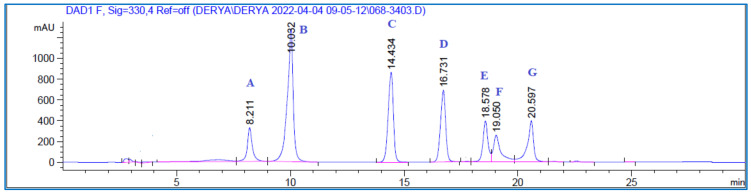
HPLC chromatograms of the standards, **A**: Chlorogenic acid, **B**: Caffeic acid, **C**: Coumaric acid, **D**: Ferulic acid, **E**: Rutin, **F**: Hyperoside, and **G**: Rosmarinic acid.

**Figure 2 antioxidants-11-01852-f002:**
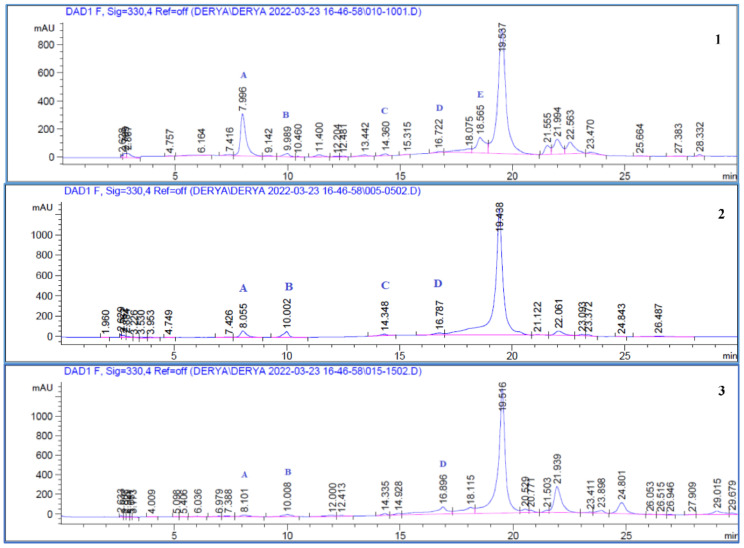
HPLC chromatograms of ethyl acetate fractions of *Arctium minus* extracts: **1**. Ethyl acetate fractions of leaves, **2**. ethyl acetate fractions of flower heads, **3**. ethyl acetate fractions of roots. Peaks: **A**—Chlorogenic acid, **B**—Caffeic acid, **C**—Coumaric acid, **D**—Ferulic acid, and **E**—Rutin.

**Table 1 antioxidants-11-01852-t001:** Total phenol and flavonoid content and quantitative determination of caffeic acid, chlorogenic acid, coumaric acid, ferulic acid, and rutin in fractions of *Arctium minus* (*n* = 3).

Extracts	Total Phenol [mg_GAE_/g_extact_]	Total Flavonoid [mg_CA_/g_extract_]	Chlorogenic Acid (% ± SD *)	Caffeic Acid (% ± SD *)	Coumaric Acid (% ± SD *)	Ferulic Acid (% ± SD *)	Rutin (% ± SD *)
**Arc L MeOH**	165.68 ± 11.22	44.41 ± 2.23	2.904 ± 0.127	ND *	ND *	ND *	0.241 ± 0.023
**Arc L CH_2_Cl_2_**	131.74 ± 4.65	44.32 ± 0.64	2.512 ± 0.005	0.035 ± 0.0002	ND *	0.002 ± 0.001	2.621 ± 0.076
**Arc L EtOAc**	318.09 ± 3.25	128.99 ± 14.00	**8.855 ± 0.175**	0.095 ± 0.004	0.016 ± 0.004	0.011 ± 0.005	**8.359 ± 0.125**
**Arc L BuOH**	208.26 ± 7.34	65.53 ± 8.89	**8.608 ± 0.292**	ND *	ND *	ND *	3.607 ± 0.059
**Arc L Aqua**	120.01 ± 2.82	33.76 ± 1.29	1.500 ± 0.007	ND *	ND *	ND *	ND *
**Arc F MeOH**	156.73 ± 7.42	43.55 ± 3.94	1.049 ± 0.081	0.029 ± 0.0005	ND *	0.017 ± 0.010	ND *
**Arc F CH_2_Cl_2_**	164.75 ± 2.44	47.58 ± 0.64	0.256 ± 0.003	0.052 ± 0.0002	ND *	0.014 ± 0.004	ND *
**Arc F EtOAc**	240.03 ± 5.95	115.85 ± 16.04	1.858 ± 0.083	0.181 ± 0.007	0.013 ± 0.013	0.206 ± 0.189	ND *
**Arc F BuOH**	194.99 ± 3.74	63.65 ± 7.59	5.001 ± 0.260	0.027 ± 0.001	ND *	0.011 ± 0.005	ND
**Arc F Aqua**	99.65 ± 2.82	24.92 ± 1.57	0.299 ± 0.005	ND *	ND *	ND *	ND *
**Arc R MeOH**	130.50 ± 11.26	30.50 ± 1.43	0.275 ± 0.089	0.023 ± 0.0001	ND *	ND *	ND *
**Arc R CH_2_Cl_2_**	142.54 ± 2.44	52.31 ± 8.63	0.132 ± 0.002	0.016 ± 0.0006	ND *	0.016 ± 0.002	ND *
**Arc R EtOAc**	**364.37 ± 7.18**	**158.56 ± 12.87**	0.519±0.021	0.102±0.005	ND *	1.001±0.113	ND *
**Arc R BuOH**	173.75± 5.95	50.51±5.41	0.506±0.015	0.030±0.0004	ND *	ND *	ND *
**Arc R Aqua**	89.47±2.13	23.88±0.14	0.162±0.003	0.021±0.0002	ND *	ND *	ND *

Values are given as mean ± standard deviation, * SD: Standard Deviation, ND: not detected, *n* = 3. Arc L MeOH = methanol extract of *A. minus* leaf, Arc L CH_2_Cl_2_ = dichloromethane extract of *A. minus* leaf, Arc L EtOAc = ethyl acetate extract of *A. minus* leaf, Arc L BuOH = butanol extract of *A. minus* leaf, Arc L Aqua = water extracts of *A. minus* leaf, Arc F MeOH = methanol extract of *A. minus* flower head, Arc F CH_2_Cl_2_ = dichloromethane extract of *A. minus* flower head, Arc F EtOAc = ethyl acetate extract of *A. minus* flower head, Arc F BuOH = butanol extract of *A. minus* flower head, Arc F Aqua = water extracts of *A. minus* flower head, Arc R MeOH = methanol extract of *A. minus* root, Arc R CH_2_Cl_2_ = dichloromethane extract of *A. minus* root, Arc R EtOAc = ethyl acetate extract of *A. minus* root, Arc R BuOH = butanol extract of *A. minus* root, Arc R Aqua = water extract of *A. minus* root.

**Table 2 antioxidants-11-01852-t002:** Calibration values for standards.

Standards	Calibration Range (μg/mL)	Linear Equation	Correlation Factor (r^2^ ± SD *)	LOD (μg/mL)	LOQ (μg/mL)
**Caffeic acid**	5–200	y = 145.14x − 85.413	0.997 ± 0.003	0.027	0.091
**Chlorogenic acid**	5–200	y = 16.82x − 48.155	0.993 ± 0.004	0.121	0.403
**Coumaric acid**	5–200	y = 109.33x + 178.3	0.999 ± 0.0001	0.135	0.450
**Ferulic acid**	5–200	y = 87.646x + 38.14	0.999 ± 0.0001	0.068	0.227
**Rutin**	5–200	y = 9.7686x + 157.04	0.991 ± 0.001	0.076	0.255

* SD: Standard Deviation.

**Table 3 antioxidants-11-01852-t003:** Precision data of the method.

Standards	Amount (μg/mL)	Intra-Day Precision (RSD *%)	Inter-Day Precision (RSD *%)
**Caffeic acid**	5 50 200	2.017 1.798 0.514	2.204 1.816 0.515
**Chlorogenic acid**	5 50 200	2.755 0.245 0.910	1.135 0.209 0.871
**Coumaric acid**	5 50 200	0.443 0.861 1.048	0.497 0.870 1.051
**Ferulic acid**	5 50 200	0.908 0.974 1.042	0.971 0.980 1.044
**Rutin**	5 50 200	0.730 3.422 0.428	0.676 3.398 0.427

* RSD: Relative standard deviation.

**Table 4 antioxidants-11-01852-t004:** Recovery assay statistical data (*n* = 3).

Standards	Concentration in Sample (mg/mL)	Amount Spiked (mg/mL)	Mean Amount Found in Mixture (mg/mL)	Mean Recovery (% ± SD *)
**Caffeic acid**	0.003	0.0015 0.003 0.006	0.0025 0.003 0.0045	101.357 ± 1.498 100.613 ± 0.787 100.351 ± 0.321
**Chlorogenic acid**	0.3	0.15 0.3 0.6	0.225 0.3 0.45	96.307 ± 3.589 101.337 ± 1.990 100.658 ± 1.704
**Coumaric acid**	0.0008	0.0004 0.0008 0.0016	0.0006 0.0008 0.0012	99.127 ± 2.538 102.453 ± 2.826 104.788 ± 2.317
**Ferulic acid**	0.0005	0.00025 0.0005 0.001	0.000375 0.0005 0.00075	103.252 ± 3.524 99.314 ± 2.409 101.117 ± 1.512
**Rutin**	0.3	0.15 0.3 0.6	0.225 0.3 0.45	98.659 ± 3.617 101.840 ± 2.378 100.243 ± 0.225

* SD: Standard deviation.

**Table 5 antioxidants-11-01852-t005:** Antioxidant activities of extracts from *A. minus*.

Extracts	DPPH	ABTS	β-Carotene
	EC50 (mg/mL)	TEAC (mmol/L Trolox)	AAC
Arc L MeOH	0.060 ± 0.004 ^a,b^	1.546 ± 0.16 ^c,d,e^ (0.5 mg/mL) 0.923 ± 0.09 ^1,2^ (0.25 mg/mL)	1422,47 ± 76.85 ^d^ (30. min) 911.53 ± 50.63 ^4^ (60. min) 667.11 ± 69.89 ^v^ (90. min)
Arc L CH_2_Cl_2_	1.196 ± 0.004 ^c^	1.08 ± 0.07 ^a,b^ (0.5 mg/mL) 0.852 ± 0.03 ^1,2^ (0.25 mg/mL)	1161.86 ± 38.64 ^a,b,c^ (30. min) 571.55 ± 36.02 ^1,2^ (60. min) 314.58 ± 49.65 ^ı^ (90 min)
Arc L EtOAc	0.019 ± 0.001 ^a^	2.041 ± 0,12 ^g^ (0.5 mg/mL) 1.425 ± 0.05 ^3^ (0.25 mg/mL)	1193.86 ± 91.66 ^a,b,c,d^ (30. min) 666.53 ± 67.40 ^1,2,3^ (60. min) 538.95 ± 84.11 ^ıı,ııı,ıv^ (90. min)
Arc L BuOH	0.048 ± 0.001 ^a,b^	1.54 ± 0.1 ^c,d,e^ (0.5 mg/mL) 1.099 ± 0.08 ^1,2,3^ (0.25 mg/mL)	1204.12 ± 49.73 ^a,b,c,d^ (30. dk) 635.32 ± 54.49 ^1,2,3^ (60. min) 487.95 ± 42.69 ^ı,ıı,ııı^ (90. min)
Arc L Aqua	0.383 ± 0.014 ^d^	1.526 ± 0.14 ^c,d,e^ (0.5 mg/mL^)^ 1.131 ± 0.08 ^1,2,3^ (0.25 mg/mL)	1232.36 ± 101.37 ^a,b,c,d^ (30. min) 795.13 ± 46.93 ^3,4^ (60. min) 581.38 ± 86.34 ^ııı,ıv,v^ (90. min)
Arc F MeOH	0.088 ± 0.004 ^b^	1.718 ± 0.1 ^e,f^ (0.5 mg/mL) 1.513 ± 0.06 ^3^ (0.25 mg/mL)	1350.89 ± 73.55 ^b,c,d^ (30. min) 807.67 ± 58.41 ^4^ (60. min) 627.37 ± 76.94 ^v^ (90. min)
Arc F CH_2_Cl_2_	1.019 ± 0.014 ^e^	1.641 ± 0.09 ^d,e,f^ (0.5 mg/mL) 1.469 ± 0.08 ^3^ (0.25 mg/mL^)^	1324.03 ± 88.61 ^b,c,d^ (30. min) 849.81 ± 68.04 ^2,3,4^ (60. min) 681.21 ± 35.80 ^ıı,ııı,ıv^ (90 min)
Arc F EtOAc	0.005 ± 0.001 ^a^	2.575 ± 0.1 ^h^ (0.5 mg/mL) 2.512 ± 0.1 ^4^ (0.25 mg/mL)	1397.47 ± 88.61 ^c,d^ (30. min) 849.81 ± 68.04 ^4^ (60. min) 681.21 ± 35.80 ^v^ (90. min)
Arc F BuOH	0.043 ± 0.003 ^a,b^	1.373 ± 0.05 ^b,c,d^ (0.5 mg/mL) 0.857 ± 0.02 ^1,2^ (0.25 mg/mL)	1223.33 ± 105.85 ^a,b,c,d^ (30. min) 648.42 ± 69.39 ^1,2,3^ (60. min) 441.45 ± 29.94 ^ı,ıı^ (90. min)
Arc F Aqua	0.383 ± 0.014 ^d^	1.040 ± 0.06 ^a^ (0.5 mg/mL) 0.77 ± 0.02 ^1^ (0.25 mg/mL)	1158.45 ± 84.29 ^a,b,c,d^ (30. min) 596.40 ± 66.81 ^1,2^ (60. min) 378.75 ± 31.43 ^ı^ (90. min)
Arc R MeOH	0.454 ± 0.018 ^f^	1.327 ± 0.08 ^a,b^ (0.5 mg/mL) 0.950 ± 0.03^1,2^ (0.25 mg/mL)	1076.80 ± 111.41 ^a,b^ (30. min) 562.84 ± 71.79 ^1^ (60. min) 357.83 ± 29.98 ^ı^ (90. min)
Arc R CH_2_Cl_2_	0.630 ± 0.056 ^g^	1.349 ± 0.07 ^a,b,c^ (0.5 mg/mL) 1.181 ± 0.06 ^1,2,3^ (0.25 mg/mL)	1177.45 ± 131.27 ^a,b,c,d^ (30. min) 638.01 ± 57.17 ^1,2,3^ (60. min) 430.19 ± 84.33 ^ı,ıı^ (90. min)
Arc R EtOAc	0.042 ± 0.00 ^a,b^	2.51 ± 0.09 ^h^ (0.5 mg/mL) 2.289 ± 0.01^4^ (0.25 mg/mL)	1350.77 ± 138.64 ^b,c,d^ (30. min) 853.87 ± 87.73 ^4^ (60. min) 657.57 ± 102.74 ^ıv,v^ (90. min)
Arc R BuOH	1.126 ± 0.019 ^h^	1.737 ± 0.06 f ^g^ (0.5 mg/mL) 1.249 ± 0.04 ^2,3^(0.25 mg/mL)	1112.40 ± 133.77 ^a,b,c^ (30. min) 592.89 ± 82.68 ^1,2^ (60. min) 468.84 ± 69.53 ^ı,ıı^ (90. min)
Arc R Aqua	0.711 ± 0.049 ^ı^	1.199 ± 0.08 ^a^ (0.5 mg/mL) 0.845 ± 0.02 ^1,2^ (0.25 mg/mL)	986.98 ± 189.82 ^a,d^ (30. min) 590.26 ± 44.39 ^1,2^ (60. min) 380.08 ± 22.10 ^ı^ (90. min)
BHA	0.007 ± 0.001 ^a^	2.34 ± 0.17 ^h^ (0.5 mg/mL) 1.23 ± 0.06 ^2,3^ (0.25 mg/mL)	1353.47 ± 45.57 ^b,c,d^ (30. min) 1201.41 ± 35.54 ^5^ (60. min) 1085.59 ± 65.75 ^vı^ (90. min)
BHT	0.008 ± 0.001 ^a^	1.89 ± 0.08 ^f,g^ (0.5 mg/mL) 1.23 ± 0.06 ^2,3^ (0.25 mg/mL)	1479.26 ± 130.89 ^d^ (30. min) 1213.43 ± 55.55 ^5^ (60. min) 1148.99 ± 42.38 ^vı^ (90. min)

Values expressed as mean  ±  standard error (*n*  =  3), statistical analysis by Tukey comparison test. Bars with the same lower case letters (a–g), numbers (1–4), and symbols (ı-vı) are not significantly (*p*  >  0.05) different. Arc L MeOH = methanol extract of *A. minus* leaf, Arc L CH_2_Cl_2_ = dichloromethane extract of *A. minus* leaf, Arc L EtOAc = ethyl acetate extract of *A. minus* leaf, Arc L BuOH = butanol extract of *A. minus* leaf, Arc L Aqua = water extracts of *A. minus* leaf, Arc F MeOH = methanol extract of *A. minus* flower head, Arc F CH_2_Cl_2_ = dichloromethane extract of *A. minus* flower head, Arc F EtOAc = ethyl acetate extract of *A. minus* flower head, Arc F BuOH = butanol extract of *A. minus* flower head, Arc F Aqua = water extracts of *A. minus* flower head, Arc R MeOH = methanol extract of *A. minus* root, Arc R CH_2_Cl_2_ = dichloromethane extract of *A. minus* root, Arc R EtOAc = ethyl acetate extract of *A. minus* root, Arc R BuOH = butanol extract of *A. minus* root, Arc R Aqua = water extract of *A. minus* root.

**Table 6 antioxidants-11-01852-t006:** Enzyme inhibitory activities of the extracts from *A. minus*.

Test Material	α-Glucosidase	α-Amylase	Lipoxygenase		Tyrosinase		IC50 (µg/mL) ± SE. *
	Inhibition % ± SE *						
	1 mg/mL	1 mg/mL	100 µg/mL	500 µg/mL	100 µg/mL	500 µg/mL	
Arc L MeOH	3.32 ± 9.80 ^b^	12.65 ± 6.40 ^a^	-	-	19.61 ± 0.45 ^e^	99.53 ± 0.2 ^a^	401.67 ± 1.69
Arc L CH_2_C_l2_	87.12 ± 8.06 ^a^	28.84 ± 5.57 ^b^	-	-	27.99 ± 0.39 ^c^	28.05 ± 0.3 ^g^	
Arc L EtOAc	-	-	-	-	57.78 ± 0.51 ^a^	99.65 ± 0.3 ^a^	93.00 ± 1.54
Arc L BuOH	24.49 ± 15.92 ^c^	30.50 ± 8.35 ^b^	-	2.92 ± 0.27 ^c^	19.46 ± 1.05 ^e^	50.76 ± 0.7 ^d^	
Arc L Aqua	15.51 ± 6.96 ^c^	5.74 ± 5.95 ^a^	-	6.67 ± 0.27 ^a^	6.35 ± 0.59 ^h^	7.12 ± 0.02 ^i^	
Arc F MeOH	-	-	-		13.32 ± 0.51 ^f,g^	20.10 ± 0.8 ^h^	
Arc F CH_2_Cl_2_	21.68 ± 3.12 ^c^	-	-	-	-	29.33 ± 0.9 ^g^	
Arc F EtOAc	40.69 ± 6.90 ^d^	-	-	-	-	-	
Arc F BuOH	6.40 ± 4.45 ^b^	-	-	-	23.50 ± 1.04 ^d^	86.68 ± 0.0 ^c^	176.67 ± 0.47
Arc F Aqua	13.32 ± 2.22 ^c^	-	-	3.59 ± 0.44 ^b,c^	-	-	
Arc R MeOH		-	-		14.89 ± 0.12 ^f^	34.18 ± 0.5 ^f^	
Arc R CH_2_Cl_2_	68.01 ± 7.02 ^a^	-	-	-	28.38 ± 1.13 ^c^	52.39 ± 0.5 ^d^	
Arc R EtOAc	36.11 ± 10.68 ^d^	-	-	-	43.26 ± 0.79 ^b^	99.40 ± 0.3 ^a^	218.75 ± 0.83
Arc R BuOH	-	-	-	4.77 ± 0.44 ^b^	10.90 ± 0.64 ^g^	34.08 ± 0.6 ^f^	
Arc R Aqua	30.40 ± 8.50 ^d^	-	-	-	-	-	-
Acarbose	79.91 ± 3.11 ^a^ (1 mg/mL)	78.4 ± 3.67 ^c^ (0.1 mg/mL)					
NDGA (20 µg/mL)			99.19 ± 0.58				
Kojic acid (500 µg/mL)						95.98 ± 0.3 ^b^	30.00 ± 0.05

* Values expressed as mean  ±  standard error (*n*  =  3), Data are presented as mean values ± 95% confidence interval. Values within a column with different superscripts differ significantly (*p* < 0.05). Arc L MeOH = methanol extract of *A. minus* leaf, Arc L CH_2_Cl_2_ = dichloromethane extract of *A. minus* leaf, Arc L EtOAc = ethyl acetate extract of *A. minus* leaf, Arc L BuOH = butanol extract of *A. minus* leaf, Arc L Aqua = water extracts of *A. minus* leaf, Arc F MeOH = methanol extract of *A. minus* flower head, Arc F CH_2_Cl_2_ = dichloromethane extract of *A. minus* flower head, Arc F EtOAc = ethyl acetate extract of *A. minus* flower head, Arc F BuOH = butanol extract of *A. minus* flower head, Arc F Aqua = water extracts of *A. minus* flower head, Arc R MeOH = methanol extract of *A. minus* root, Arc R CH_2_Cl_2_ = dichloromethane extract of *A. minus* root, Arc R EtOAc = ethyl acetate extract of *A. minus* root, Arc R BuOH = utanol extract of *A. minus* root, Arc R Aqua = water extract of *A. minus* root.

**Table 7 antioxidants-11-01852-t007:** IC50 values (µg/mL) of extracts in breast cancer cell lines.

IC50 (µg/mL)		
Extracts	MCF-7	MDA-MB-217
Arc L MeOH	>500	43.87 ± 3.40 ^d^
Arc L CH_2_ Cl_2_	64.90 ± 6.83 ^e^	21.39 ± 2.43 ^b^
Arc L EtOAc	>125	30.05 ± 4.44 ^b,c^
Arc L BuOH	>500	>125
Arc L Aqua	>1000	>125
Arc F MeOH	>250	71.68 ± 3.11 ^f^
Arc F CH_2_Cl_2_	39.65 ± 3.21 ^c^	10.80 ± 1.26 ^a^
Arc F EtOAc	>125	>125
Arc F BuOH	41.67 ± 3.76 ^c,d^	27.75 ± 8.95 ^b^
Arc F Aqua	>125	>125
Arc R MeOH	>1000	>250
Arc R CH_2_Cl_2_	46.72 ± 0.98 ^d^	13.41 ± 2.37 ^a^
Arc R EtOAc	>250	75.69 ± 2.43 ^f^
Arc R BuOH	>1000	>250
Arc R Aqua	>1000	>1000

Values (µg/mL) given as mean ± standard error (*n* = 3). Bars with the same lower case letters (a–f) are not significantly (*p*  >  0.05) different. Arc L MeOH = methanol extract of *A. minus* leaf, Arc L CH_2_Cl_2_ = dichloromethane extract of *A. minus* leaf, Arc L EtOAc = ethyl acetate extract of *A. minus* leaf, Arc L BuOH = butanol extract of *A. minus* leaf, Arc L Aqua = water extracts of *A. minus* leaf, Arc F MeOH = methanol extract of *A. minus* flower head, Arc F CH_2_Cl_2 =_ dichloromethane extract of *A. minus* flower head, Arc F EtOAc = ethyl acetate extract of *A. minus* flower head, Arc F BuOH = butanol extract of *A. minus* flower head, Arc F Aqua = water extracts of *A. minus* flower head, Arc R MeOH = methanol extract of *A. minus* root, Arc R CH_2_Cl_2_ = dichloromethane extract of *A. minus* root, Arc R EtOAc = ethyl acetate extract of *A. minus* root, Arc R BuOH = butanol extract of *A. minus* root, Arc R Aqua = water extract of *A. minus* root.

## Data Availability

The data presented in this study are available in the article.

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
