# Peer review of "Phytochemical Composition and Biological Activities of Arctium minus (Hill) Bernh.: A Potential Candidate as Antioxidant, Enzyme Inhibitor, and Cytotoxic Agent"

_antioxidants, 2022, doi:10.3390/antiox11101852_

Round 1

Reviewer 1 Report

Overall, the manuscript needs significant changes before it could be accepted for publication. My comments are below.

  • The abstract must include data regarding the critical finds by the authors in terms of data of important findings. 
  • The introduction must have a clear hypothesis and significantly develop the second paragraph of this manuscript.
  • Overall there is the repetition of the information which could be avoided.  
  •  Check figure ligands; they are carelessly written.
  • Discussion should include more information and references related to the relevant and related works. 
  • Restructure and carefully edit the conclusion section.

Author Response

  • The abstract must include data regarding the critical finds by the authors in terms of data of important findings

Data for key findings are included in the summary.

  • The introduction must have a clear hypothesis and significantly develop the second paragraph of this manuscript

For the introduction part, the hypothesis has been stated more clearly and paragraph 2 has been rewritten.

  • Overall there is the repetition of the information which could be avoided.

In general, parts that are considered to be repetitive information have been revised.

  • Check figure ligands; they are carelessly written.

Figure ligands are checked and corrected.

  • Discussion should include more information and references related to the relevant and related works.

The discussion was revised and enriched with related studies.

  • Restructure and carefully edit the conclusion section.

Result section has been rearranged.

Reviewer 2 Report

The manuscript can be of interest to wide readers of journals and contributes to existing knowledge on the subject matter. However, I have pointed out few pertinent points for improving the clarity of the content and boosting the scientific soundness of the manuscript.

Too generalized statements might be omitted

 Authors need to present findings as integral values or percentage

 Line 43: What do you mean by “the plant is very active”?

Line 72-76: Include citations for the statements.

Line 86-89: Include citations for the statements.

Line 90-92: Include citations for the statements.

Line 53 and 65: No need to define abbreviations twice

“reactive oxygen species (ROS)”

Line 68: Define

“BRCA mutation”

Materials and methods

Information on study coordinates must be added.

Line 135: Include the make, model and address for the equipment used in the experiments

C18 column, HPLC, Spectrophotometer,

Line 266: ND: not determined,-- or not detected? If it is not determined, then why not determined?

Table 1, 2,3,4,7: Statisitical analysis is missing

Discussions

Poor discussions.

Authors need to strengthen the discussion section by adding more interpretations of recorded findings supported by peer-findings, include more citations from recent works and publications.

Include the mechanism.

3.2. Antioxidant Activity

Include more citations

Author Response

  • Too generalized statements might be omitted.

Generalized statements has been checked and revised again.

  • Authors need to present findings as integral values or percentage

Findings given as integral values or percentage

  • Line 43: What do you mean by “the plant is very active”?

The phrase "plant is very active" in Line 43 has been changed.

  • Line 72-76: Include citations for the statements.

Reference has been added

  • Line 86-89: Include citations for the statements.

Reference has been added

  • Line 90-92: Include citations for the statements.

Reference has been added

  • Line 53 and 65: No need to define abbreviations twice “reactive oxygen species (ROS)

Define of abbreviations have been corrected.

  • Line 68: Define “BRCA mutation”

The sentence in which this expression is used has been removed from the article.

Materials and methods

  • Information on study coordinates must be added.

Study coordinates have been added.

  • Line 135: Include the make, model and address for the equipment used in the experiments:C18 column, HPLC, Spectrophotometer,

Models for the equipment have been added.

  • Line 266: ND: not determined,-- or not detected? If it is not determined, then why not determined?

“Not determined” statement has been corrected  as “not detected”

  • Table 1, 2,3,4,7: Statisitical analysis is missing

Statisitical analyses is completed for table 7 and the necessary information for tables 2, 3, and 4 are given separately under the heading of statistical analysis in the material method. In addition, statistical analysis was not performed as no comparison was made with any standard substance in the experiments specified in Table 1.

Discussions

  • Poor discussions.

              Authors need to strengthen the discussion section by adding more interpretations of recorded findings supported by peer-findings, include more citations from recent works and publications. Include the mechanism.

Discussion section has been strengthened with new literatüre data.

  • 2. Antioxidant Activity

Include more citations

More references for the antioxidant section has been added and expanded the discussion section.

Reviewer 3 Report

The manuscript entitled " Phytochemical Composition and Biological Activities of Arctium minus (Hill) Bernh.: A Potential Candidate as Antioxidant, Enzyme Inhibitor and Cytotoxic Agent" mainly focuses on use of plant extract enzyme inhibition and cytotoxic effects. From the contents of the manuscript to estimate, this research is good in the qualitative and quantitative analysis. In my judgement, it needs to be revised for publication in this journal.

  Abstract

Abstract is well written. However, results should be further specified specifically in quantitative values.

Introduction

Paragraph 1st need further description specifically in regards to medicinal plants its uses and treatments of disorders. 

The author must discuss recent advances of the use of botanicals and plant extracts for antioxidant, enzyme inhibition and cytotoxic effects.

Also discuss antioxidant and enzyme inhibition mechanism of plant extracts.

Origin and distribution of the studied plant must be added.

Also add significance of the objectives.

Discussion must be compared with recent studies section wise.

Also add recommendations in the conclusion section

Author Response

  • Abstract is well written. However, results should be further specified specifically in quantitative values.

              In the summary, the results are stated in quantitative values.

               It has been corrected as stated by the referee.

Paragraph 1st need further description specifically in regards to medicinal plants its uses and treatments of disorders.

Description about medicinal plants has been added.

  • The author must discuss recent advances of the use of botanicals and plant extracts for antioxidant, enzyme inhibition and cytotoxic effects.

Current studies have been tried to be included.

  • Also discuss antioxidant and enzyme inhibition mechanism of plant extracts.

Necessary information has been added to both the discussion and the introduction part.

  • Origin and distribution of the studied plant must be added.

The information about the origin and distribution of plant is given in the introduction.

  • Also add significance of the objectives.

It has been added.

  • Discussion must be compared with recent studies section wise.

Recent studies has been added.

  • Also add recommendations in the conclusion section

Recommendations has been added in the conclusion section

Round 2

Reviewer 1 Report

Authors have significantly improved the manuscript. Therefore, it can be accepted for publication.

Author Response

English of the manuscript was checked by a native speaker. 

Reviewer 2 Report

There are some minor issues. For instances

1. The author required to use citations for  the following sentences

line: 498-500

Line:537-538

Line 156:  Include the make, model and address for the equipment used in the experiments:   C18 column, HPLC, Spectrophotometer,

Author Response

The author required to use citations for the following sentences

Reference has been used for line: 498-500

Reference has been used for line:537-538

Line 156:  Include the make, model and address for the equipment used in the experiments:   C18 column, HPLC, Spectrophotometer,

Model and address have been addeed  for the equipment.